# Association of coincident self-reported mental health problems and alcohol intake with all-cause and cardiovascular disease mortality: A Norwegian pooled population analysis

Eirik Degerud[1]*, Gudrun Høiseth[2,3,4], Jørg Mørland[1,4], Inger Ariansen[1], Sidsel Graff-Iversen[1], Eivind Ystrom[1,5,6], Luisa Zuccolo[7], Øyvind Næss[1,8]

1 Norwegian Institute of Public Health, Oslo, Norway, 2 Center for Psychopharmacology, Diakonhjemmet Hospital, Oslo, Norway, 3 Department of Forensic Medicine, Oslo University Hospital, Oslo, Norway, 4 Institute of Clinical Medicine, University of Oslo, Oslo, Norway, 5 PROMENTA Research Center, Department of Psychology, University of Oslo, Oslo, Norway, 6 PharmacoEpidemiology and Drug Safety Research Group, School of Pharmacy, University of Oslo, Oslo, Norway, 7 MRC Integrative Epidemiology Unit, Department of Population Health Sciences, Bristol Medical School, University of Bristol, Bristol, United Kingdom, 8 Institute of Health and Society, University of Oslo, Oslo, Norway

* eide@fhi.no

**Data Availability Statement:** The results in this study are based on de-identified data from human

## Abstract

### Background

The disease burden attributable to mental health problems and to excess or harmful alcohol use is considerable. Despite a strong relationship between these 2 important factors in population health, there are few studies quantifying the mortality risk associated with their co-occurrence in the general population. The aim of this study was therefore to investigate cardiovascular disease (CVD) and all-cause mortality according to self-reported mental health problems and alcohol intake in the general population.

### Methods and findings

We followed 243,372 participants in Norwegian health surveys (1994–2002) through 2014 for all-cause and CVD mortality by data linkage to national registries. The mean (SD) age at the time of participation in the survey was 43.9 (10.6) years, and 47.8% were men. During a mean (SD) follow-up period of 16.7 (3.2) years, 6,587 participants died from CVD, and 21,376 died from all causes. Cox models estimated hazard ratios (HRs) with 95% CIs according to a mental health index (low, 1.00–1.50; high, 2.01–4.00; low score is favourable) based on the General Health Questionnaire and the Hopkins Symptom Checklist, and according to self-reported alcohol intake (low, <2; light, 2–11.99; moderate, 12–23.99; high, ≥24 grams/day). HRs were adjusted for age, sex, educational level, marital status, and CVD risk factors. Compared to a reference group with low mental health index score and low alcohol intake, HRs (95% CIs) for all-cause mortality were 0.93 (0.89, 0.97; $p = 0.001$), 1.00 (0.92, 1.09; $p = 0.926$), and 1.14 (0.96, 1.35; $p = 0.119$) for low index score combined with light, moderate, and high alcohol intake, respectively. HRs (95% CIs) were 1.22 (1.14,

research participants. Data can be made available to all interested researchers upon request. For more information, please contact the Norwegian National Institute of Public health, Division of Health data and Digitalisation (lhu@fhi.no).

**Funding:** This study is part of a larger research project. The project received funding from the Research Council of Norway (Grant Number 2137788) https://www.forskningsradet.no/en/. The recipient was the project leader ØN. The funders had no role in study design, data collection and analysis, decision to publish, or preparation of the manuscript.

**Competing interests:** The authors have declared that no competing interests exist.

**Abbreviations:** CONOR, Cohort of Norway; CVD, cardiovascular disease; HDL-C, high-density lipoprotein cholesterol; HR, hazard ratio.

1.31; $p < 0.001$), 1.24 (1.15, 1.33; $p < 0.001$), 1.43 (1.23, 1.66; $p < 0.001$), and 2.29 (1.87, 2.80; $p < 0.001$) for high index score combined with low, light, moderate, and high alcohol intake, respectively. For CVD mortality, HRs (95% CIs) were 0.93 (0.86, 1.00; $p = 0.058$), 0.90 (0.76, 1.07; $p = 0.225$), and 0.95 (0.67, 1.33; $p = 0.760$) for a low index score combined with light, moderate, and high alcohol intake, respectively, and 1.11 (0.98, 1.25; $p = 0.102$), 0.97 (0.83, 1.13; $p = 0.689$), 1.01 (0.71, 1.44; $p = 0.956$), and 1.78 (1.14, 2.78; $p = 0.011$) for high index score combined with low, light, moderate, and high alcohol intake, respectively. HRs for the combination of a high index score and high alcohol intake (HRs: 2.29 for all-cause and 1.78 for CVD mortality) were 64% (95% CI 53%, 74%; $p < 0.001$) and 69% (95% CI 42%, 97%; $p < 0.001$) higher than expected for all-cause mortality and CVD mortality, respectively, under the assumption of a multiplicative interaction structure. A limitation of our study is that the findings were based on average reported intake of alcohol without accounting for the drinking pattern.

## Conclusions

In the general population, the mortality rates associated with more mental health problems and a high alcohol intake were increased when the risk factors occurred together.

---

### Author summary

#### Why was this study done?

- Many people experience negative health outcomes because of alcohol use or because of mental health problems.

- Many people both drink alcohol and experience mental health problems, but we do not have much data showing whether the combination of drinking alcohol and mental health problems is associated with additional negative health consequences.

#### What did the researchers do and find?

- We grouped people sampled from the Norwegian adult general population according to their self-reported levels of mental health problems and alcohol intake to compare their risk of all-cause and cardiovascular disease mortality.

- The risk of all-cause and cardiovascular disease mortality was higher among people with more mental health problems, as scored on a mental health index, and a high alcohol intake ($\geq$24 grams/day) than would be expected for the linear combination of having high alcohol intake and a high score on the mental health index.

#### What do these findings mean?

- The findings suggest that co-occurring alcohol intake and mental health problems are associated with increased negative health effects including all-cause and cardiovascular-disease-related mortality.

- Our findings may help to inform clinical recommendations and low-risk drinking guidelines regarding potential risks of alcohol use by individuals with mental health problems.

- The findings warrant future studies with longitudinal data that can shed more light on the mechanisms underlying the interaction between alcohol intake, mental health, and mortality.

## Introduction

Excessive and harmful alcohol consumption is associated with violence and accidents and chronic diseases such as cancer and cardiovascular disease (CVD), and is a leading contributor to the disease burden in many countries [1]. Alcohol drinking guidelines inform the public about levels and patterns of drinking that are less harmful, as judged by estimates of average risk in the general population [2,3]. They also give reasons to avoid alcohol, such as in specific groups where the health consequences of alcohol could be worse than in the general population [4,5].

Guidelines in both Canada and the UK mention mental health problems as a reason to avoid alcohol [2,3]. The recommendation is based on evidence of a positive association between alcohol intake or alcohol use disorder and mental health problems [6–9], and there is a possibility that mental health problems could worsen because of drinking. The disease burden attributed to mental health problems is considerable [10–14]. Excess or harmful alcohol use could have a negative influence on mental health by interfering with social relationships and environments that underlie mental well-being, such as family relations and employment [6], or through direct biological mechanisms that reduce grey and white matter brain volume or interfere with neurotransmitter functioning [15]. On the other hand, there is evidence suggesting that acute or chronic symptoms of mental distress might lead to an increase in alcohol intake [16,17], supporting the 'self-medication theory'. In addition, a positive association between alcohol use and mental health problems could also arise or be explained in part by genetic or environmental factors influencing both alcohol use and mental health [18,19].

A bi-directional relationship between mental health and alcohol intake, or shared genetic or environmental factors, could lead to negative feedback cycles [8] that over time contribute to more mental health problems, heavier alcohol or psychotropic drug use, poor diet, physical inactivity, and a less protective social and socioeconomic environment. The consequence could be an increased mortality risk among individuals in the general population who both experience mental health problems and drink alcohol, but few studies have quantified this relationship [20]. We provide relevant data from a large pooled sample of population-based cardiovascular health surveys in Norway, focusing on the quantity of alcohol intake and intensity of mental health problems reported by people in the general population. The first objective was to estimate the risk of mortality from all causes and CVD according to self-reported alcohol intake and according to self-reported mental health problems separately. The second objective was to investigate whether risks of all-cause and CVD-related mortality are increased in persons who report mental health problems and alcohol intake in combination, which would suggest an interaction between mental health problems and alcohol intake.

## Methods

### Study population selection, data linkage, and ethical approval

The Age 40 Program is a collection of cardiovascular health surveys that invited all men and women aged 40–42 years living in 13 out of 19 counties in Norway to participate. The Cohort

of Norway (CONOR) is a collection of population-based cardiovascular health surveys from different geographical areas in Norway, both rural and urban, where various age groups were invited [21]. To conduct this study, we identified surveys within the Age 40 Program and CONOR that measured both alcohol intake and mental health problems. The identified surveys were performed between 1994 and 2002, had a response rate ranging from 38% to 78%, and constitute a source population of 307,541 visits, where both sexes are rather equally represented and where approximately 73% were in the age range 35 to 50 years at the time of participation in the survey (S1 Table). Use of the term 'visits' reflects the fact that a small proportion of individuals attended more than 1 survey. The vast majority, however, attended only a single survey and were represented by a single visit. The overlap was coincidental, with the exception of 2 surveys from the Tromsø study, where a large proportion of the participants in the fifth survey were previous participants in the fourth survey.

When selecting the study population, we only allowed each individual to be represented by a single visit. For the small proportion with multiple visits, we choose the first for individuals who attended the fourth and fifth survey of the Tromsø study ($n$ = 7,017). For individuals who coincidentally attended a survey in CONOR and in the Age 40 Program, we placed an arbitrary priority on CONOR, but chose the visit from the Age 40 Program if data on drinking status were missing in CONOR. Individuals with missing or inconsistent survey data on alcohol, mental health, or covariates were excluded, as well as individuals missing covariate or outcome data obtained by registry linkage (S1 Fig).

This study is based on person-level data from health surveys and national registries. The personal identification number (PIN) that is unique to each Norwegian resident ensures nearly complete linkage. Statistics Norway received data from the data sources, replaced the PIN with a dummy ID number, and sent the data to the authors. The authors could then link the data. The Regional Committee for Medical and Health Research Ethics South-East (11/1676) approved the study and gave exemption regarding consent in the surveys where this was not obtained. This study is reported as per the Reporting of Studies Conducted Using Observational Routinely-Collected Health Data (RECORD) guideline, which is an extension of the Strengthening the Reporting of Observational Studies in Epidemiology (STROBE) guideline (S1 Checklist). The study is part of a larger research project with a prospective protocol (S1 Protocol). In S1 Text, we provide an overview of differences between the conducted study and the planned study as described in the protocol. Data cleaning, harmonisation of survey data, selection of the study population, and statistical analyses were performed using R statistical software (S2 Text).

## Mental health index

A mental health index modified from the General Health Questionnaire [22] and the Hopkins Symptom Checklist [23] assessed self-reported mental health in the surveys (S3 Text). The index correlates with the Hopkins Symptom Checklist and the Hospital Anxiety and Depression Scale [24]. It is based on 7 questions that inquire whether the person in the past 2 weeks ever felt 'nervous and unsettled', 'troubled by anxiety', 'secure and calm', 'irritable', 'happy and optimistic', 'sad/depressed', or 'lonely'. We scored the 4 possible answers ('no' = 1, 'a little' = 2, 'quite a bit' = 3, and 'very' = 4), and, by reversing the scores for the questions 'secure and calm' and 'happy and optimistic', we obtained for each individual a sum score (range 7–28) and a mean score per question (1–4), in which higher scores indicate more problems. To increase sample size, we replaced a single missing value by the sample mean value for that question. Participants missing 2 or more values were excluded. The index was categorised as follows based on the mean score per question: low, 1.00–1.50; medium, 1.51–2.00; and high, 2.01–4.00.

## Alcohol consumption

Questions relevant to alcohol intake differed between surveys (S1 Table). Only a few surveys provided data on lifetime abstaining, and we therefore defined lifetime and current abstainers collectively as current abstainers. We estimated the average intake of alcohol (grams/day) among current drinkers using 2 methods depending on the data available in each survey. The first method used the question 'How many glasses of beer, wine, or spirits do you usually drink during a two-week period?', which was answered separately for each beverage type (range 0–50, higher values truncated). For individuals with data on at least 1 beverage type, the number of glasses were added together and the sum divided by 14. Conversion from glasses to grams of alcohol was performed using the following definitions: 1 litre of pure alcohol is 789 grams, a glass of beer is 33.3 cl and 4.5% alcohol (11.8 grams), a glass of wine is 15 cl and 12% alcohol (14 grams), and a glass of liquor is 4 cl and 40% alcohol (12.6 grams). The second method combined drinking frequency obtained from the question 'Approximately how often during the past year have you consumed alcohol?' ('A few times last year' = 6 times/year, 'Approximately 1 time a month' = 12 times/year, '2–3 times a month' = 30 times/year, 'Approximately 1 time a week' = 52 times/year, '2–3 times a week' = 130 times/year, and '4–7 times a week' = 286 times/year), with the reported number of glasses consumed per occasion (0–20, higher values truncated) obtained from the question 'When you drank alcohol, how many glasses did you usually drink?' The calculated number of glasses per year was divided by 365 and converted to grams of alcohol (1 glass = 12.8 grams of pure alcohol). Alcohol intake was categorised as follows: current abstainers; low, <2 grams/day; light, 2–11.99 grams/day; moderate, 12–23.99 grams/day; and high, ≥24 grams/day.

## Covariates

Marital status at survey participation (married, divorced, separated/widowed, never married) was ascertained using the National Registry. The National Education Database provided data on the highest level of attained education (range 1–8; 1 = primary school and 8 = master's degree or higher). Questionnaires provided data on smoking status, level of physical activity (range 1–4), history of diabetes, history of CVD, and family history of coronary heart disease. In some surveys, questionnaires also provide data on the use of antidepressants and tranquillizers in the last 4 weeks (yes/no) and whether the person has ever sought help for psychological problems (yes/no). Smoking status was defined as never, light former, heavy former, light current, and heavy current smoker (light and heavy defined by ≤20 and >20 pack years). We obtained systolic blood pressure (mm Hg), resting heart rate (bpm), and body mass index (BMI, kg/m$^2$) from objective measurements performed by survey personnel. Serum triglycerides (mmol/l), total cholesterol, and high-density lipoprotein cholesterol (HDL-C) were obtained using biochemical measurements in non-fasting blood samples taken by study personnel.

## Outcome

Participants were followed until emigration (until December 31, 2012), death, or December 31, 2014. Emigration status was obtained from the National Registry, and mortality data from the Norwegian Cause of Death Registry. Cause of death is based on certificates filled out by on-site medical doctors, and occasionally (4.3% in 2015) on autopsies [25]. The 9th (1994–95) and 10th (1996–2014) revision of the International Classification of Diseases (ICD) were used to identify deaths from all causes and from CVD (ICD–9: 390–459; ICD–10: I00–I99).

## Statistical analysis

We first described the distribution of covariate values according to categories of the mental health index and alcohol intake. Analysis of variance and the chi-squared test were used to assess differences between groups. Next, we visualised the crude association of alcohol intake with the mental health index, and assessed the age- and sex-adjusted association using ordinary linear regression. The first objective was to assess the separate associations of mental health problems and of alcohol intake with all-cause and CVD mortality. To address this, we used Cox proportional hazard models to estimate hazard ratios (HRs), 95% CIs, and $p$-values according to the mental health index and according to alcohol intake (separate analysis). Time on study was used as the timescale. To address the second objective of a potential interaction between mental health and alcohol intake, we present HRs for their association within strata of each other (stratified analysis) and for their joint association by using a common reference category (joint analysis). This setup provides the information required to interpret the interaction from different perspectives, to assess the influence from covariates, and to perform recalculations [26,27]. For the separate, joint, and stratified analyses, we fitted an unadjusted Cox model, a model adjusted for age and sex, and a model further adjusted for smoking status, education, marital status, history of CVD, BMI, heart rate, physical activity, diabetes, family history of coronary heart disease, serum cholesterol, and serum triglycerides. Age, BMI, serum cholesterol, and serum triglycerides were fitted as linear variables, and the other variables were fitted as categorical variables. These covariates were included because they could confound associations involving alcohol intake and mental health. We left out HDL-C (only available for a smaller sample) and blood pressure because they were more likely to play a mediating role in associations involving alcohol intake. The decision to exclude blood pressure did not materially alter risk estimates in analyses involving mental health.

In the separate, stratified, and joint analyses, we estimated HRs using categorical exposure variables, with a low mean score on the mental health index (1.00–1.50) and a low alcohol intake ($<$2 grams/day) as the single or combined reference category, where applicable. Based on HRs from the joint analysis, we tested for interaction by assessing whether the observed joint HR differed from the expected joint HR using a multiplicative interaction structure. We did this for all possible combinations of the exposure categories. Specifically, we divided the observed joint HR for a given combination (e.g., moderate index score + moderate alcohol intake) by the expected joint HR (e.g., the product of the HRs from moderate index score + low alcohol intake and low index score + moderate alcohol intake). Standard errors and 95% CIs were obtained using the Delta method. In the separate and stratified analyses, we also estimated HRs per unit increase of the exposure variables on a continuous scale (for average alcohol intake among current drinkers only), used interaction terms (to test for a difference in slope) as a complementary test for interaction, and present the associations graphically to support the interpretation of the associations. The functional form of the associations was obtained by including the continuous exposure variables as penalised smoothed splines in Cox models adjusted for age and sex. The regression terms (log HR) were plotted as a function of the mental health index and alcohol intake. The function is centred (log HR = 0) at the mean value of the predictors.

Lastly, we used ordinary linear regression to assess whether self-reporting of alcohol intake was consistent with alcohol intake as judged by HDL-C [28]. HDL-C was regressed on age, sex, and average alcohol intake (current drinkers only), and the results are presented as regression coefficients with 95% CIs. Analyses were repeated among men and women with different scores on the mental health index to assess the possibility of differential bias in self-reporting, together with interaction terms to test for a difference in slope.

## Results

### Study population

The source population comprised 295,126 participants, of whom 12,415 had participated in more than 1 of the health surveys. Among these individuals with overlapping participation, we selected 1 visit. Next, we excluded in total 51,754 (17.5%) participants for missing values: Participants were excluded for missing drinking status ($n$ = 6,655), leaving 288,471 individuals with current drinking status, and for missing values on the mental health index ($n$ = 28,278), alcohol intake among current drinkers ($n$ = 8,713), follow-up data ($n$ = 21), or covariate data ($n$ = 8,087). The number of participants available for statistical analyses was 243,372 (S1 Fig).

Potentially eligible individuals excluded for missing values were on average older, more likely to be male, and more likely to have died from any cause and CVD during follow-up than individuals in the source population (S2 Table). As a result, the proportion of people dying among complete cases in the study population was somewhat lower than in the source population.

### Participant characteristics

Participant characteristics (Tables 1 and S3) were unevenly distributed according to the mental health index and alcohol intake. Participants with higher scores on the mental health index were more often women, had less education on average, and were more likely to have ever been married and to have a history of diabetes and CVD. They also had less favourable levels of CVD risk factors including more drinking and current smoking, less physical activity, higher heart rate, and higher serum triglycerides. Participants with a higher score on the mental health index were more likely to have used antidepressants and tranquillizers in the last 4 weeks, as well as to have sought help for psychological problems, within the subsets where this information was assessed. Participants who drank more alcohol were more often male, more likely to have ever been married, had higher serum HDL-C and triglycerides on average, and to a large extent had also attained more education and reported more physical activity on average. Current abstainers were less likely to be a current or previous smoker than current drinkers. Among current drinkers, alcohol intake was positively associated with current light and with current and previous heavy smoking. We observed a non-linear distribution pattern for age, systolic blood pressure, diabetes, and history of CVD. Drinkers reporting light (2–11.99 grams/day) and moderate (12–23.99 grams/day) alcohol intake were younger and had lower blood pressure, less diabetes, and less previous CVD in comparison with current abstainers and drinkers with low intake (<2 grams/day) and high intake (≥24 grams/day). Current abstainers and drinkers with a high alcohol intake were more likely to have used antidepressants or tranquillizers during the past 4 weeks, as well as more likely to have ever sought help for psychological problems, than drinkers reporting low, light, and moderate intake of alcohol.

### Alcohol intake and the mental health index

The crude relationship between alcohol intake and the mental health index appeared non-linear (S2 Fig; Table 1). The analysis adjusted for age and sex showed that the mean score on the mental health index was, in comparison with people drinking <2 grams/day, slightly lower and thus more favourable among participants with a light alcohol intake, slightly higher among current abstainers and participants with a moderate intake, and considerably higher among participants with a high intake. The regression coefficients (95% CIs) were 0.033 (0.027, 0.039; $p < 0.001$) among current abstainers and −0.011 (−0.015, −0.007; $p < 0.001$), 0.051 (0.044, 0.059; $p < 0.001$), and 0.187 (0.171, 0.202; $p < 0.001$) among participants with

**Table 1. Descriptive statistics according to self-reported mental health problems and alcohol intake in the study population.**

| Characteristic | Mental health index (mean score) | n | All participants (n = 243,372) | Current abstainers (n = 22,496) | Average intake of alcohol (grams/day) | | | | p-Value |
|---|---|---|---|---|---|---|---|---|---|
| | | | | | Low (<2 grams/day) (n = 85,961) | Light (2–11.99 grams/day) (n = 116,170) | Moderate (12–23.99 grams/day) (n = 15,944) | High (≥24 grams/day) (n = 2,801) | |
| Sex (male) | All | 243,372 | 116,218 (47.8%) | 7,942 (35.3%) | 32,930 (38.3%) | 60,847 (52.4%) | 12,110 (76.0%) | 2,389 (85.3%) | <0.001 |
| | 1.00–1.50 | 148,428 | 73,888 (49.8%) | 4,917 (37.3%) | 21,336 (40.7%) | 39,375 (54.4%) | 7,088 (77.4%) | 1,172 (87.0%) | <0.001 |
| | 1.51–2.00 | 71,546 | 32,759 (45.8%) | 2,164 (33.5%) | 8,889 (35.6%) | 17,120 (50.1%) | 3,799 (75.4%) | 787 (84.7%) | <0.001 |
| | 2.01–4.00 | 23,398 | 9,571 (40.9%) | 861 (30.0%) | 2,705 (31.6%) | 4,352 (44.9%) | 1,223 (69.8%) | 430 (81.9%) | <0.001 |
| | p-Value | | <0.001 | <0.001 | <0.001 | <0.001 | <0.001 | 0.016 | |
| Age | All | 243,372 | 43.9 (10.6) | 49.0 (14.2) | 44.6 (11.4) | 42.6 (9.0) | 42.8 (9.1) | 43.9 (10.4) | <0.001 |
| | 1.00–1.50 | 148,428 | 43.9 (10.5) | 48.8 (14.0) | 44.5 (11.4) | 42.6 (8.9) | 43.0 (9.3) | 44.3 (10.6) | <0.001 |
| | 1.51–2.00 | 71,546 | 43.9 (10.7) | 49.3 (14.3) | 44.7 (11.5) | 42.5 (8.9) | 42.6 (8.9) | 44.0 (10.5) | <0.001 |
| | 2.01–4.00 | 23,398 | 44.4 (11.1) | 49.8 (14.6) | 44.9 (11.6) | 42.9 (9.3) | 42.7 (8.8) | 42.8 (9.2) | <0.001 |
| | p-Value | | <0.001 | <0.001 | <0.001 | 0.442 | 0.032 | 0.007 | |
| Mental health index (1–4) | All | 243,372 | 1.51 (0.42) | 1.54 (0.48) | 1.51 (0.43) | 1.49 (0.40) | 1.54 (0.44) | 1.67 (0.55) | 0.012 |
| | 1.00–1.50 | 148,428 | 1.25 (0.15) | 1.24 (0.15) | 1.25 (0.15) | 1.25 (0.15) | 1.26 (0.15) | 1.26 (0.15) | <0.001 |
| | 1.51–2.00 | 71,546 | 1.72 (0.15) | 1.73 (0.15) | 1.72 (0.15) | 1.71 (0.15) | 1.72 (0.15) | 1.73 (0.15) | 0.002 |
| | 2.01–4.00 | 23,398 | 2.48 (0.37) | 2.53 (0.40) | 2.49 (0.37) | 2.46 (0.35) | 2.49 (0.36) | 2.61 (0.44) | 0.007 |
| | p-Value | | <0.001 | <0.001 | <0.001 | <0.001 | <0.001 | <0.001 | |
| Average alcohol intake (grams/day) | All | 220,876 | 4.64 (5.81) | — | 0.57 (0.71) | 5.38 (2.57) | 15.9 (3.12) | 34.6 (12.8) | <0.001 |
| | 1.00–1.50 | 135,254 | 4.48 (5.34) | — | 0.58 (0.71) | 5.34 (2.55) | 15.8 (3.09) | 32.9 (10.8) | <0.001 |
| | 1.51–2.00 | 65,091 | 4.79 (5.94) | — | 0.57 (0.70) | 5.44 (2.59) | 15.9 (3.11) | 34.2 (11.5) | <0.001 |
| | 2.01–4.00 | 20,531 | 5.20 (7.90) | — | 0.51 (0.68) | 5.48 (2.64) | 16.2 (3.23) | 39.9 (17.6) | <0.001 |
| | p-Value | | <0.001 | — | <0.001 | <0.001 | <0.001 | <0.001 | |
| Education (1–8) | All | 243,372 | 4.05 (1.64) | 3.67 (1.66) | 3.82 (1.58) | 4.23 (1.63) | 4.44 (1.68) | 4.31 (1.75) | <0.001 |
| | 1.00–1.50 | 148,428 | 4.08 (1.63) | 3.78 (1.67) | 3.84 (1.57) | 4.25 (1.62) | 4.49 (1.66) | 4.49 (1.71) | <0.001 |
| | 1.51–2.00 | 71,546 | 4.08 (1.66) | 3.64 (1.68) | 3.85 (1.60) | 4.26 (1.65) | 4.45 (1.69) | 4.35 (1.77) | <0.001 |
| | 2.01–4.00 | 23,398 | 3.75 (1.64) | 3.21 (1.54) | 3.60 (1.58) | 3.98 (1.64) | 4.13 (1.74) | 3.78 (1.74) | <0.001 |
| | p-Value | | <0.001 | <0.001 | <0.001 | <0.001 | <0.001 | <0.001 | |
| Ever married | All | 243,372 | 47,686 (19.6%) | 3,353 (14.9%) | 15,899 (18.5%) | 23,310 (20.1%) | 4,216 (26.4%) | 908 (32.4%) | <0.001 |
| | 1.00–1.50 | 148,428 | 27,161 (18.3%) | 1,782 (13.5%) | 9,230 (17.6%) | 13,579 (18.8%) | 2,185 (23.9%) | 385 (28.6%) | <0.001 |
| | 1.51–2.00 | 71,546 | 15,061 (21.1%) | 1,033 (16.0%) | 4,900 (19.6%) | 7,399 (21.7%) | 1,420 (28.2%) | 309 (33.3%) | <0.001 |
| | 2.01–4.00 | 23,398 | 5,464 (23.4%) | 538 (18.8%) | 1,769 (20.7%) | 2,332 (24.1%) | 611 (34.9%) | 214 (40.8%) | <0.001 |
| | p-Value | | <0.001 | <0.001 | <0.001 | <0.001 | <0.001 | <0.001 | |
| Current smoker | All | 243,372 | 83,359 (34.3%) | 3,635 (16.2%) | 28,425 (33.1%) | 42,698 (36.8%) | 7,153 (44.9%) | 1,448 (51.7%) | <0.001 |
| | 1.00–1.50 | 148,428 | 47,112 (31.7%) | 1,696 (12.9%) | 16,216 (30.9%) | 24,854 (34.4%) | 3,735 (40.8%) | 611 (45.4%) | <0.001 |
| | 1.51–2.00 | 71,546 | 25,426 (35.5%) | 1,070 (16.6%) | 8,473 (34.0%) | 13,018 (38.1%) | 2,388 (47.4%) | 477 (51.3%) | <0.001 |
| | 2.01–4.00 | 23,398 | 10,821 (46.2%) | 869 (30.3%) | 3,736 (43.6%) | 4,826 (49.8%) | 1,030 (58.8%) | 360 (68.6%) | <0.001 |
| | p-Value | | <0.001 | <0.001 | <0.001 | <0.001 | <0.001 | <0.001 | |

(*Continued*)

**Table 1.** (Continued)

| Characteristic | Mental health index (mean score) | n | All participants (n = 243,372) | Current abstainers (n = 22,496) | Average intake of alcohol (grams/day) | | | | p-Value |
| --- | --- | --- | --- | --- | --- | --- | --- | --- | --- |
| | | | | | Low (<2 grams/day) (n = 85,961) | Light (2–11.99 grams/day) (n = 116,170) | Moderate (12–23.99 grams/day) (n = 15,944) | High (≥24 grams/day) (n = 2,801) | |
| Physical activity (1–4) | All | 243,372 | 2.04 (0.94) | 1.80 (0.92) | 1.95 (0.93) | 2.13 (0.94) | 2.19 (0.96) | 2.08 (0.96) | <0.001 |
| | 1.00–1.50 | 148,428 | 2.09 (0.95) | 1.87 (0.94) | 2.00 (0.95) | 2.18 (0.95) | 2.25 (0.96) | 2.16 (0.98) | <0.001 |
| | 1.51–2.00 | 71,546 | 2.00 (0.92) | 1.74 (0.89) | 1.91 (0.91) | 2.08 (0.92) | 2.14 (0.94) | 2.07 (0.94) | <0.001 |
| | 2.01–4.00 | 23,398 | 1.85 (0.91) | 1.62 (0.87) | 1.77 (0.88) | 1.96 (0.91) | 2.01 (0.96) | 1.87 (0.93) | <0.001 |
| | p-Value | | <0.001 | <0.001 | <0.001 | <0.001 | <0.001 | <0.001 | |
| BMI (kg/m²) | All | 243,372 | 25.7 (3.90) | 26.3 (4.52) | 25.8 (4.18) | 25.4 (3.59) | 25.8 (3.50) | 26.0 (3.70) | <0.001 |
| | 1.00–1.50 | 148,428 | 25.7 (3.81) | 26.3 (4.40) | 25.8 (4.06) | 25.5 (3.52) | 25.9 (3.47) | 26.2 (3.58) | <0.001 |
| | 1.51–2.00 | 71,546 | 25.6 (3.95) | 26.3 (4.55) | 25.8 (4.25) | 25.3 (3.64) | 25.7 (3.47) | 26.1 (3.77) | <0.001 |
| | 2.01–4.00 | 23,398 | 25.8 (4.33) | 26.7 (4.92) | 26.0 (4.62) | 25.4 (3.93) | 25.6 (3.75) | 25.5 (3.83) | <0.001 |
| | p-Value | | 0.590 | <0.001 | 0.055 | <0.001 | <0.001 | <0.001 | |
| Systolic blood pressure (mm Hg) | All | 243,372 | 130.1 (17.4) | 134.4 (21.7) | 130.1 (18.1) | 128.9 (16.1) | 132.2 (15.7) | 134.4 (16.1) | <0.001 |
| | 1.00–1.50 | 148,428 | 130.5 (17.4) | 134.6 (21.4) | 130.6 (18.0) | 129.3 (16.1) | 132.6 (15.8) | 135.1 (16.1) | <0.001 |
| | 1.51–2.00 | 71,546 | 129.6 (17.5) | 134.3 (22.0) | 129.6 (18.3) | 128.3 (16.0) | 131.7 (15.7) | 133.9 (16.1) | <0.001 |
| | 2.01–4.00 | 23,398 | 129.2 (17.8) | 134.4 (22.4) | 128.7 (17.9) | 127.6 (16.3) | 131.2 (15.3) | 133.8 (15.8) | <0.001 |
| | p-Value | | <0.001 | <0.001 | <0.001 | <0.001 | <0.001 | 0.063 | |
| Use of antidepressants in the last 4 weeks (yes) | All | 27,028 | 1,008 (3.7%) | 78 (7.4%) | 385 (4.1%) | 401 (3.0%) | 96 (3.8%) | 48 (7.4%) | <0.001 |
| | 1.00–1.50 | 14,785 | 189 (1.3%) | 12 (2.2%) | 73 (1.4%) | 86 (1.1%) | 12 (0.9%) | 6 (2.0%) | 0.072 |
| | 1.51–2.00 | 8,956 | 333 (3.7%) | 19 (6.1%) | 137 (4.4%) | 128 (2.8%) | 32 (3.8%) | 17 (7.9%) | <0.001 |
| | 2.01–4.00 | 3,287 | 486 (14.8%) | 47 (23.5%) | 175 (15.3%) | 187 (12.8%) | 52 (15.4%) | 25 (17.9%) | 0.001 |
| | p-Value | | <0.001 | <0.001 | <0.001 | <0.001 | <0.001 | <0.001 | |
| Use of tranquillizers in the last 4 weeks (yes) | All | 27,069 | 1,206 (4.5%) | 100 (9.4%) | 441 (4.7%) | 486 (3.6%) | 119 (4.7%) | 60 (9.3%) | <0.001 |
| | 1.00–1.50 | 14,785 | 212 (1.4%) | 22 (4.1%) | 83 (1.6%) | 81 (1.1%) | 20 (1.5%) | 6 (2.0%) | <0.001 |
| | 1.51–2.00 | 8,975 | 402 (4.5%) | 26 (8.3%) | 150 (4.9%) | 171 (3.8%) | 37 (4.4%) | 18 (8.4%) | <0.001 |
| | 2.01–4.00 | 3,309 | 592 (17.9%) | 52 (25.9%) | 208 (17.9%) | 234 (15.9%) | 62 (18.1%) | 36 (26.7%) | 0.001 |
| | p-Value | | <0.001 | <0.001 | <0.001 | <0.001 | <0.001 | <0.001 | |
| Ever sought help for a psychological problem (yes) | All | 108,744 | 12,531 (11.5%) | 1,443 (15.6%) | 4,737 (11.8%) | 5,249 (10.2%) | 847 (12.4%) | 255 (18.9%) | <0.001 |
| | 1.00–1.50 | 65,292 | 3,384 (5.2%) | 360 (6.7%) | 1,328 (5.5%) | 1,467 (4.7%) | 182 (4.8%) | 47 (7.3%) | <0.001 |
| | 1.51–2.00 | 32,362 | 4,674 (14.4%) | 485 (18.2%) | 1,746 (14.9%) | 2,039 (13.3%) | 329 (14.9%) | 75 (17.3%) | <0.001 |
| | 2.01–4.00 | 11,090 | 4,473 (40.3%) | 598 (47.3%) | 1,663 (40.2%) | 1,743 (37.9%) | 336 (41.3%) | 133 (47.8%) | <0.001 |
| | p-Value | | <0.001 | <0.001 | <0.001 | <0.001 | <0.001 | <0.001 | |

Values are presented as mean (standard deviation) or count (percentage). Group differences were assessed with analysis of variance and the chi-squared test. Mental health index is presented as the mean score per question (range 1–4) on the 7 questions that constitute the mental health index, where a high score indicates a greater number of self-reported mental health issues. We present attained education and physical activity at the mean values of 8 and 4 ordinal categories, respectively, where a higher mean value indicates higher education and higher activity level.

light, moderate, and high average alcohol intake, respectively, in comparison with a low intake.

## Mental health and the risk of all-cause and CVD mortality

The mean (SD) follow-up time in the study population was 16.7 (3.2) years, with a minimum of 0 days and a maximum of 20.3 years. In total, 21,376 participants died and 6,587 died from CVD in this period. The visualisation of the associations between the mental health index and

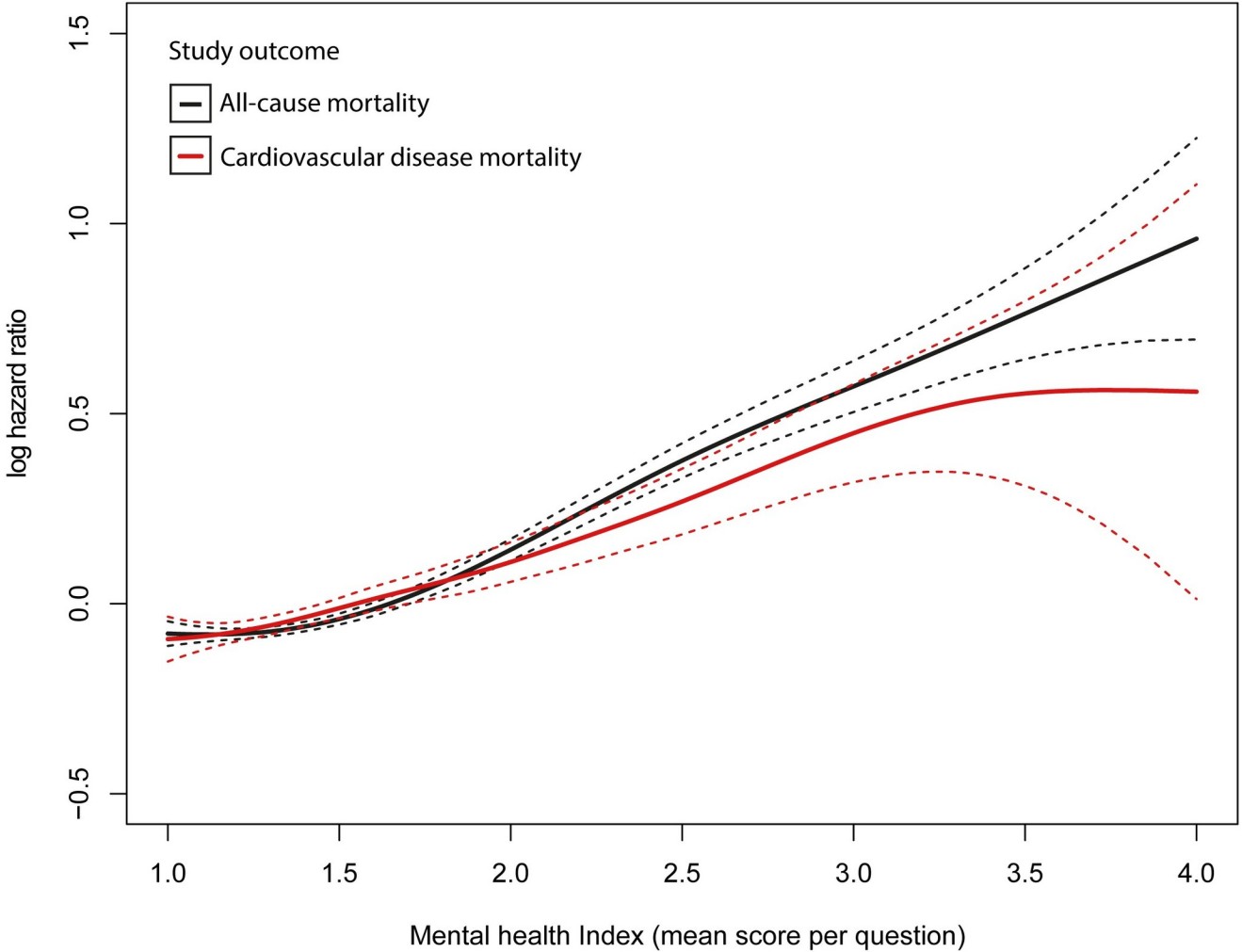

**Fig 1. Visualisation of the association between mental health problems (mental health index, mean score per question, range 1–4) and all-cause and cardiovascular disease mortality among all participants ($n$ = 243,372).** The mental health index was fitted as a continuous variable using penalised smoothed splines in a Cox model adjusted for age and sex. The vertical axis shows the log hazard ratio (the exponential value is equivalent to the hazard ratio). The solid line depicts the spline and the dashed curves depict twice the standard error (as plotted by the termplot function in R statistical software, which was used for visualisation). Twice the standard error is a close approximation of a 95% confidence interval, as 95% of the values fall within $1.96 \times$ standard error of the mean if the values are normally distributed. The curve is centred (log hazard ratio = 0) at the mean value of the predictors (mean score: 1.51). The $y$-axis is fixed between −0.5 and 1.5.

all-cause and CVD mortality supported a linear relationship with both outcomes (Fig 1). In formal analyses (S4 Table), the age- and sex-adjusted HRs (95% CIs) per unit increase in the mean score per question on the mental health index were 1.35 (1.32, 1.39; $p < 0.001$) for all-cause mortality and 1.29 (1.22, 1.36; $p < 0.001$) for CVD mortality. These estimates were attenuated to 1.17 (1.14, 1.21; $p < 0.001$) and 1.07 (1.01, 1.13; $p = 0.020$) in multivariable adjusted analysis, respectively.

## Alcohol intake and the risk of all-cause and CVD mortality

In comparison with drinkers with a low alcohol intake, the multivariable adjusted HRs (95% CIs) for all-cause mortality were 1.19 (1.15, 1.24; $p < 0.001$) among current abstainers, and 0.93 (0.90, 0.96; $p < 0.001$), 1.03 (0.97, 1.09; $p = 0.386$), and 1.33 (1.19, 1.48; $p < 0.001$), respectively, among drinkers reporting light, moderate, and high alcohol intake (S5 Table). Multivariable adjusted HRs (95% CIs) for CVD mortality were, in comparison with drinkers with a low alcohol intake, 1.25 (1.18, 1.34; $p < 0.001$) among current abstainers and 0.90 (0.85, 0.96; $p = 0.001$), 0.93 (0.82, 1.05; $p = 0.257$), and 1.04 (0.83, 1.31; $p = 0.724$), respectively, among drinkers reporting light, moderate, and high alcohol intake.

## Interaction between mental health problems and alcohol intake regarding all-cause and CVD mortality

To assess the joint association of mental health problems and alcohol intake with all-cause (Table 2) and CVD (Table 3) mortality, we used the participants with a low mental health index score (1.00–1.50 mean score) and low alcohol intake (<2 grams/day) as a common reference category. The observed HRs (95% CIs) for all-cause mortality in categories combining a low score on the mental health index (1.00–1.50) with light, moderate, and high alcohol intake were 0.93 (0.89, 0.97; $p = 0.001$), 1.00 (0.92, 1.09; $p = 0.926$), and 1.14 (0.96, 1.35; $p = 0.119$), respectively. We then multiplied each of the abovementioned HRs by the HR for the category combining a high score on the mental health index (2.01–4.00) with a low alcohol intake, which was 1.22 (95% CI 1.14, 1.31; $p < 0.001$). This provides the expected joint HRs (1.13, 1.22, and 1.39, respectively) for the categories combining a high score on the mental health index with light, moderate, and high alcohol intake, assuming no interaction on the multiplicative scale. The observed joint HRs for these categories, however, were 1.24 (95% CI 1.15, 1.33; $p < 0.001$), 1.43 (95% CI 1.23, 1.66; $p < 0.001$), and 2.29 (95% CI 1.87, 2.80; $p < 0.001$), respectively, which were 1.09 (95% CI 1.03, 1.15; $p = 0.002$), 1.17 (95% CI 1.06, 1.27; $p = 0.001$), and 1.64 (95% CI 1.53, 1.74; $p < 0.001$) higher than the expected joint HRs, suggesting interaction. In addition, for the combined category of high alcohol intake and a medium score on the mental health index (1.51–2.00), the observed joint HR for all-cause mortality was 1.33 (95% CI 1.11, 1.60; $p = 0.003$), which was 1.13 (95% CI 1.00, 1.36; $p = 0.019$) higher than the expected HR of 1.17 (derived from $1.14 \times 1.03$). The observed joint HR for CVD mortality was also higher than expected for a high alcohol intake combined with a high mean score on the mental health index, but not for other combinations. Observed and expected joint HRs were 1.78 (95% CI 1.14, 2.78; $p = 0.011$) and 1.05 (derived from $0.95 \times 1.11$), respectively, while the HR for interaction was 1.69 (95% CI 1.42, 1.97; $p < 0.001$).

In stratified analyses adjusted for age and sex, a 1-unit increase in mean score per question on the mental health index was consistently associated with a higher HR for both mortality outcomes at all alcohol intake levels, but only for all-cause mortality in multivariable adjusted analyses (S4 Table). The point estimates and test for slope difference indicated a more pronounced change in HRs for all-cause mortality among people reporting moderate and high alcohol intake versus low intake. Multivariable adjusted HRs for all-cause mortality per unit change in the mental health index in the low, moderate, and high alcohol intake strata were 1.15 (95% CI 1.10, 1.21; $p < 0.001$), 1.32 (95% CI 1.17, 1.48; $p < 0.001$), and 1.32 (95% CI 1.17, 1.48; $p < 0.001$), respectively. HRs for the tests for slope differences were 1.22 (95% CI 1.08, 1.38; $p = 0.002$) for moderate versus low intake and 1.38 (95% CI 1.16, 1.63; $p < 0.001$) for high versus low intake. In stratified analysis of alcohol intake, HRs for both mortality outcomes were consistently higher among current abstainers in comparison with current drinkers with a low intake. In analysis of current drinkers only, the visualisation of the associations of average alcohol intake with all-cause mortality and CVD mortality (Fig 2) indicated a more adverse

**Table 2. All-cause mortality according to mental health problems and alcohol intake using a joint reference category.**

| Measure | Mental health index (range 1–4) | Current abstainers | Average alcohol intake (grams/day) | | | |
|---|---|---|---|---|---|---|
| | | | Low (<2 grams/day) | Light (2–11.99 grams/day) | Moderate (12–23.99 grams/day) | High (≥24 grams/day) |
| **Event/no event (*n*)** | 1.00–1.50 | 2,319/10,855 | 4,959/47,480 | 4,223/68,092 | 634/8,519 | 142/1,205 |
| | 1.51–2.00 | 1,298/5,157 | 2,513/22,443 | 1,954/32,214 | 360/4,678 | 117/812 |
| | 2.01–4.00 | 695/2,172 | 1,050/7,516 | 835/8,852 | 180/1,573 | 97/428 |
| **Unadjusted** | | | | | | |
| HR (95% CI) | 1.00–1.50 | 1.92 (1.83, 2.02), *p* < 0.001 | Referent | 0.62 (0.59, 0.64), *p* < 0.001 | 0.76 (0.70, 0.82), *p* < 0.001 | 1.20 (1.02, 1.42), *p* = 0.030 |
| | 1.51–2.00 | 2.24 (2.11, 2.38), *p* < 0.001 | 1.08 (1.03, 1.13), *p* = 0.001 | 0.61 (0.58, 0.64), *p* < 0.001 | 0.79 (0.71, 0.88), *p* < 0.001 | 1.47 (1.22, 1.76), *p* < 0.001 |
| | 2.01–4.00 | 2.83 (2.61, 3.06), *p* < 0.001 | 1.35 (1.26, 1.44), *p* < 0.001 | 0.95 (0.88, 1.02), *p* = 0.138 | 1.17 (1.01, 1.35), *p* = 0.042 | 2.24 (1.83, 2.73), *p* < 0.001 |
| HR for interaction (95% CI) | 1.00–1.50 | — | Referent | — | — | — |
| | 1.51–2.00 | 1.08 (1.01, 1.14), *p* = 0.013 | — | 0.91 (0.85, 0.98), *p* = 0.009 | 0.96 (0.84, 1.09), *p* = 0.550 | 1.13 (0.94, 1.32), *p* = 0.159 |
| | 2.01–4.00 | 1.09 (1.04, 1.15), *p* = 0.001 | — | 1.14 (1.09, 1.19), *p* < 0.001 | 1.14 (1.06, 1.23), *p* = 0.001 | 1.38 (1.27, 1.48), *p* < 0.001 |
| **Adjusted for age and sex** | | | | | | |
| HR (95% CIs) | 1.00–1.50 | 1.13 (1.07, 1.18), *p* < 0.001 | Referent | 0.90 (0.86, 0.93), *p* < 0.001 | 0.99 (0.91, 1.07), *p* = 0.740 | 1.20 (1.02, 1.42), *p* = 0.029 |
| | 1.51–2.00 | 1.29 (1.22, 1.38), *p* < 0.001 | 1.10 (1.04, 1.15), *p* < 0.001 | 0.94 (0.89, 0.99), *p* = 0.013 | 1.14 (1.02, 1.27), *p* = 0.020 | 1.56 (1.30, 1.88), *p* < 0.001 |
| | 2.01–4.00 | 1.70 (1.56, 1.84), *p* < 0.001 | 1.42 (1.33, 1.52), *p* < 0.001 | 1.43 (1.33, 1.54), *p* < 0.001 | 1.80 (1.55, 2.09), *p* < 0.001 | 3.68 (3.01, 4.51), *p* < 0.001 |
| HR for interaction (95% CI) | 1.00–1.50 | — | Referent | — | — | — |
| | 1.51–2.00 | 1.04 (0.99, 1.12), *p* = 0.215 | — | 0.95 (0.89, 1.02), *p* = 0.141 | 1.05 (0.94, 1.16), *p* = 0.369 | 1.18 (1.00, 1.36), *p* = 0.035 |
| | 2.01–4.00 | 1.06 (1.01, 1.11), *p* = 0.015 | — | 1.11 (1.08, 1.17), *p* < 0.001 | 1.28 (1.22, 1.35), *p* < 0.001 | 2.16 (2.09, 2.21), *p* < 0.001 |
| **Multivariable adjusted** | | | | | | |
| HR (95% CI) | 1.00–1.50 | 1.19 (1.13, 1.25), *p* < 0.001 | Referent | 0.93 (0.89, 0.97), *p* = 0.001 | 1.00 (0.92, 1.09), *p* = 0.926 | 1.14 (0.96, 1.35), *p* = 0.119 |
| | 1.51–2.00 | 1.27 (1.19, 1.35), *p* < 0.001 | 1.03 (0.98, 1.08), *p* = 0.200 | 0.93 (0.88, 0.98), *p* = 0.005 | 1.04 (0.94, 1.16), *p* = 0.440 | 1.33 (1.11, 1.60), *p* = 0.003 |
| | 2.01–4.00 | 1.39 (1.28, 1.51), *p* < 0.001 | 1.22 (1.14, 1.31), *p* < 0.001 | 1.24 (1.15, 1.33), *p* < 0.001 | 1.43 (1.23, 1.66), *p* < 0.001 | 2.29 (1.87, 2.80), *p* < 0.001 |
| HR for interaction (95% CI) | 1.00–1.50 | — | Referent | — | — | — |
| | 1.51–2.00 | 1.03 (0.99, 1.12), *p* = 0.353 | — | 0.97 (0.89, 1.01), *p* = 0.351 | 1.01 (0.94, 1.16), *p* = 0.863 | 1.13 (1.00, 1.36), *p* = 0.119 |
| | 2.01–4.00 | 0.95 (0.88, 1.03), *p* = 0.203 | — | 1.09 (1.03, 1.15), *p* = 0.002 | 1.17 (1.06, 1.27), *p* = 0.001 | 1.64 (1.53, 1.74), *p* < 0.001 |

HRs, 95% CIs, and *p*-values derived from Cox models. The multivariable model was adjusted for age, sex, education, marital status, smoking, physical activity, body mass index, resting heart rate, total cholesterol concentration, triglyceride concentration, diabetes, family history of coronary heart disease, and history of CVD. The HR for interaction assesses whether the observed joint HR differed from the expected joint HR derived using a multiplicative interaction structure, calculated by dividing the observed joint HR for a given combination (e.g., moderate index score + moderate alcohol intake) by the expected joint HR (e.g., the product of the HRs from moderate index score + low alcohol intake and low index score + moderate alcohol intake). Standard errors and 95% CIs were obtained using the Delta method. CI, confidence interval; HR, hazard ratio.

**Table 3. CVD mortality according to mental health problems and alcohol intake using a joint reference category.**

| Measure | Mental health index (range 1–4) | Current abstainers | Average alcohol intake (grams/day) | | | |
| --- | --- | --- | --- | --- | --- | --- |
| | | | Low (<2 grams/day) | Light (2–11.99 grams/day) | Moderate (12–23.99 grams/day) | High (≥24 grams/day) |
| **Event/no event (n)** | 1.00–1.50 | 898/12,276 | 1,606/50,833 | 1,148/71,167 | 153/9,000 | 34/1,313 |
| | 1.51–2.00 | 499/5,956 | 822/24,134 | 506/33,662 | 96/4,942 | 25/904 |
| | 2.01–4.00 | 256/2,611 | 308/8,258 | 184/9,503 | 32/1,721 | 20/505 |
| **Unadjusted** | | | | | | |
| HR (95% CI) | 1.00–1.50 | 2.30 (2.12, 2.49), $p < 0.001$ | Referent | 0.52 (0.48, 0.56), $p < 0.001$ | 0.56 (0.48, 0.66), $p < 0.001$ | 0.88 (0.63, 1.24), $p = 0.465$ |
| | 1.51–2.00 | 2.66 (2.40, 2.94), $p < 0.001$ | 1.09 (1.00, 1.19), $p = 0.043$ | 0.49 (0.44, 0.54), $p < 0.001$ | 0.65 (0.53, 0.79), $p < 0.001$ | 0.96 (0.65, 1.42), $p = 0.830$ |
| | 2.01–4.00 | 3.20 (2.80, 3.65), $p < 0.001$ | 1.22 (1.08, 1.38), $p = 0.002$ | 0.64 (0.55, 0.75), $p < 0.001$ | 0.63 (0.45, 0.90), $p = 0.011$ | 1.40 (0.90, 2.18), $p = 0.131$ |
| HR for interaction (95% CI) | 1.00–1.50 | — | Referent | — | — | — |
| | 1.51–2.00 | 1.06 (0.95, 1.17), $p = 0.276$ | — | 0.86 (0.73, 0.99), $p = 0.052$ | 1.06 (0.84, 1.27), $p = 0.593$ | 1.00 (0.56, 1.44), $p = 1.000$ |
| | 2.01–4.00 | 1.14 (1.04, 1.25), $p = 0.005$ | — | 1.01 (0.89, 1.15), $p = 0.888$ | 0.92 (0.64, 1.22), $p = 0.625$ | 1.30 (1.02, 1.60), $p = 0.022$ |
| **Adjusted for age and sex** | | | | | | |
| HR (95% CIs) | 1.00–1.50 | 1.25 (1.15, 1.36), $p < 0.001$ | Referent | 0.84 (0.78, 0.91), $p < 0.001$ | 0.79 (0.67, 0.94), $p = 0.006$ | 0.89 (0.63, 1.25), $p = 0.516$ |
| | 1.51–2.00 | 1.43 (1.29, 1.58), $p < 0.001$ | 1.11 (1.02, 1.21), $p = 0.013$ | 0.85 (0.77, 0.94), $p = 0.002$ | 1.03 (0.84, 1.27), $p = 0.742$ | 1.04 (0.70, 1.54), $p = 0.875$ |
| | 2.01–4.00 | 1.79 (1.57, 2.04), $p < 0.001$ | 1.31 (1.16, 1.48), $p < 0.001$ | 1.10 (0.95, 1.29), $p = 0.210$ | 1.16 (0.81, 1.64), $p = 0.418$ | 2.75 (1.77, 4.28), $p < 0.001$ |
| HR for interaction (95% CI) | 1.00–1.50 | — | Referent | — | — | — |
| | 1.51–2.00 | 1.03 (0.92, 1.14), $p = 0.601$ | — | 0.91 (0.79, 1.03), $p = 0.164$ | 1.17 (0.99, 1.36), $p = 0.052$ | 1.05 (0.63, 1.45), $p = 0.830$ |
| | 2.01–4.00 | 1.09 (0.99, 1.19), $p = 0.066$ | — | 1.09 (0.89, 1.12), $p = 0.142$ | 1.12 (0.91, 1.33), $p = 0.244$ | 2.36 (2.21, 2.49), $p < 0.001$ |
| **Multivariable adjusted** | | | | | | |
| HR (95% CI) | 1.00–1.50 | 1.25 (1.15, 1.36), $p < 0.001$ | Referent | 0.93 (0.86, 1.00), $p = 0.058$ | 0.90 (0.76, 1.07), $p = 0.225$ | 0.95 (0.67, 1.33), $p = 0.760$ |
| | 1.51–2.00 | 1.32 (1.19, 1.46), $p < 0.001$ | 1.04 (0.96, 1.13), $p = 0.340$ | 0.89 (0.80, 0.98), $p = 0.019$ | 1.02 (0.83, 1.26), $p = 0.814$ | 0.95 (0.64, 1.42), $p = 0.765$ |
| | 2.01–4.00 | 1.36 (1.19, 1.56), $p < 0.001$ | 1.11 (0.98, 1.25), $p = 0.102$ | 0.97 (0.83, 1.13), $p = 0.689$ | 1.01 (0.71, 1.44), $p = 0.956$ | 1.78 (1.14, 2.78), $p = 0.011$ |
| HR for interaction (95% CI) | 1.00–1.50 | — | Referent | — | — | — |
| | 1.51–2.00 | 1.02 (0.89, 1.14), $p = 0.767$ | — | 0.92 (0.78, 1.05), $p = 0.275$ | 1.09 (0.87, 1.32), $p = 0.426$ | 0.96 (0.45, 1.46), $p = 0.900$ |
| | 2.01–4.00 | 0.98 (0.83, 1.14), $p = 0.815$ | — | 0.94 (0.77, 1.12), $p = 0.528$ | 1.01 (0.69, 1.33), $p = 0.957$ | 1.69 (1.42, 1.97), $p < 0.001$ |

HRs, 95% CIs, and p-values derived from Cox models. The multivariable model was adjusted for age, sex, education, marital status, smoking, physical activity, body mass index, resting heart rate, total cholesterol concentration, triglyceride concentration, diabetes, family history of coronary heart disease, and history of CVD. The HR for interaction assesses whether the observed joint HR differed from the expected joint HR derived using a multiplicative interaction structure, calculated by dividing the observed joint HR for a given combination (e.g., moderate index score + moderate alcohol intake) by the expected joint HR (e.g., the product of the HRs from moderate index score + low alcohol intake and low index score + moderate alcohol intake). Standard errors and 95% CIs were obtained using the Delta method. CI, confidence interval; HR, hazard ratio.

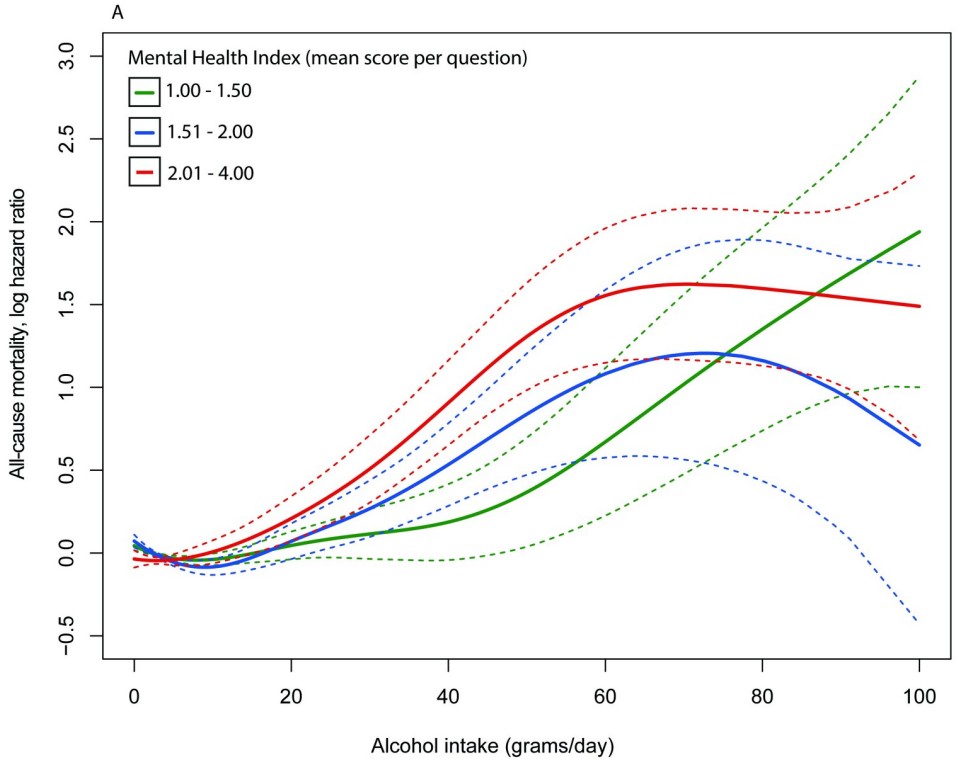

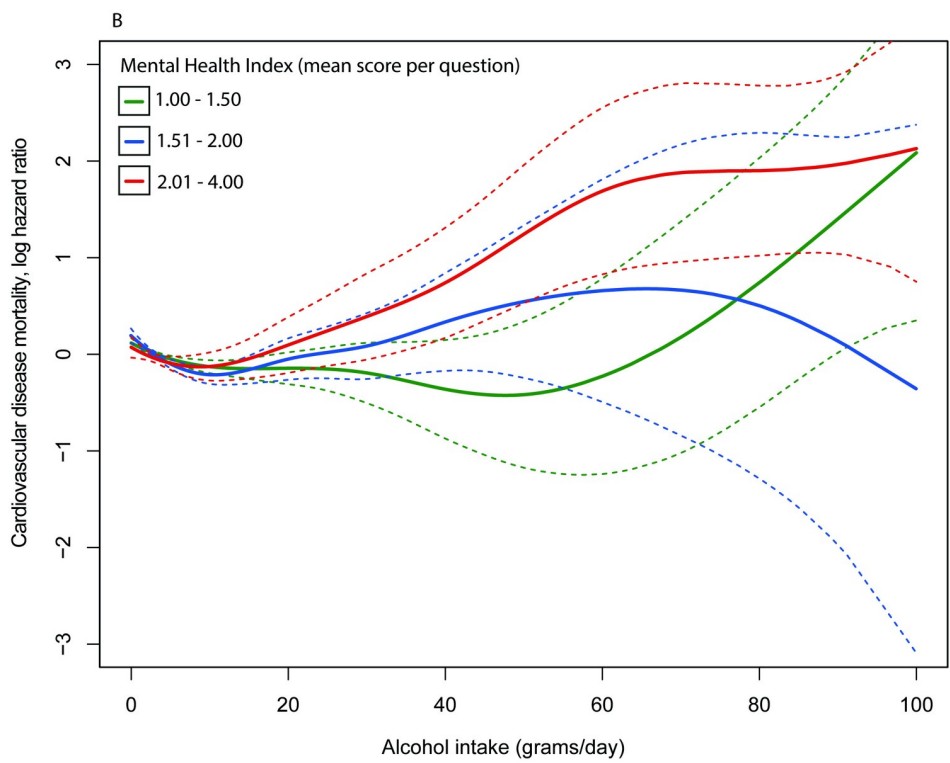

**Fig 2. Visualisation of the associations of alcohol intake (grams per day) with all-cause mortality and cardiovascular disease mortality among current drinkers ($n$ = 220,876) in strata of the mental health index.** (A) All-cause mortality; (B) cardiovascular disease mortality. Average alcohol consumption was fitted as a continuous variable using penalised smoothed splines in a Cox model adjusted for age and sex. The vertical axis shows the log hazard ratio (the exponential value is equivalent to the hazard ratio). The solid line depicts the spline and the dashed curves depict twice the standard error (as plotted by the termplot function in R statistical software, which was used for visualisation). Twice the standard error is a close approximation of a 95% confidence interval, as 95% of the values fall within $1.96 \times$ standard error of the mean if the values are normally distributed. The curve was centred (log hazard ratio = 0) at the mean value of alcohol intake in each mental health index stratum (1.00–1.50, 4.48 grams/day; 1.51–2.00, 4.79 grams/day; 2.01–4.00, 5.20 grams/day). The curves were plotted individually using a fixed $y$-axis between −0.5 and 3 and then superimposed.

association among people with more mental health problems at higher alcohol intake levels, albeit with wide CIs. HRs per unit increase in average alcohol intake were 1.06 (95% CI 1.04, 1.08; $p < 0.001$) for all-cause mortality and 1.07 (95% CI 1.02, 1.11; $p = 0.003$) for CVD mortality among people with a high mental health index score, but not different from 1 among people with a low or medium score (S5 Table). HRs derived from the tests for slope difference were 1.07 (95% CI 1.04, 1.10; $p < 0.001$) for all-cause and 1.10 (95% CI 1.04, 1.16; $p = 0.001$) for CVD mortality, indicating a more pronounced change in HR per unit change in alcohol intake among people with a high versus low score on the mental health index.

### Additional analyses

Crude distributions of HDL-C according to sex, alcohol intake, and the mental health index are presented in S3 Table. In analyses adjusted for age, the regression coefficient (95% CI) per unit increase in alcohol (1 gram/day) was 0.013 (0.013, 0.014) and 0.007 (0.006, 0.007) mmol/l among women and men, respectively. Regression coefficient (95% CI) values for low, medium, and high mental health index scores respectively were 0.014 (0.013, 0.015), 0.013 (0.012, 0.014), and 0.012 (0.010, 0.013) for women ($p$ for difference in slope with low score as reference: 0.051 and 0.001), and 0.007 (0.007, 0.007), 0.006 (0.006, 0.007), and 0.006 (0.005, 0.007) for men ($p$ for difference: 0.017 and 0.046).

## Discussion

### Brief summary

In this study of adults from the general Norwegian population, we observe that those who report more mental health problems, as judged by a high score on a mental health index, had a 26% and 10% higher risk of all-cause and CVD mortality, respectively, in comparison with people who report fewer problems. Furthermore, we find non-linear associations involving alcohol intake, with a 7% and 10% lower risk of all-cause and CVD mortality, respectively, with light intake (2–11.99 grams/day), and a 33% higher risk of all-cause mortality with high intake ($\geq$24 grams/day), in comparison with low intake (<2 grams/day). In joint analyses, we observed a 129% and 78% higher risk of all-cause and CVD mortality, respectively, for the combination of high alcohol intake and high mean score on the mental health index (compared with low alcohol intake and a low index score). These estimates were 64%–69% higher than expected under the assumption of a multiplicative interaction structure.

### Mental health and mortality

A higher score on the mental health index was associated with higher all-cause and CVD mortality. This is in agreement with existing research on the health consequences of mental health problems [10–14]. A considerable part of the age- and sex-adjusted association was accounted for by multivariable adjustment, which is in line with the interpretation that mental health

may impact mortality via changes in health behaviour. An alternative and more conservative interpretation is that the association was confounded by the risk factors in the multivariable model. Support for either interpretation would require a temporal sequence, but as mental health and other risk factors were measured at the same time, the study was limited in this regard.

## Alcohol intake and mortality

Previous studies have reported a J- or U-shaped risk pattern between alcohol intake and all-cause mortality or CVD [29,30]. A recent consortium-based individual-level analysis of current drinkers found that an intake of 14 grams/day was associated with the lowest risk of both outcomes [31]. Our data are in support of lower CVD and all-cause mortality at light alcohol intake (2–11.99 grams/day), and also numerically lower CVD mortality at moderate intake (12–23.99 grams/day), in comparison with a low intake (<2 grams/day). The data also support higher all-cause mortality for high intake (≥24 grams/day). The difference in HR between current abstainers and current drinkers with a low intake of alcohol is less likely to be the result of a difference in the reported alcohol intake, and more likely to reflect unmeasured confounding or reverse causation as suggested by genetically informed studies [32].

## Joint analysis of mental health and alcohol intake and mortality

HRs for all-cause mortality for the joint association of mental health problems and alcohol intake were 9%–64% higher than expected under an assumption of a multiplicative interaction structure. For CVD mortality, the associations were in line with a potential interaction among participants reporting the highest alcohol intake level and the most mental health problems, but this interaction did not manifest as a gradient starting at lower intake or index levels. Data in support of an interaction involving alcohol intake and mental health problems with all-cause mortality were observed in a previous study by Greenfield et al., in which 18% of the participants were defined with high levels of depression using a 20-item Centre for Epidemiological Studies Depression Scale [20]. We performed our study in a larger sample, allowing us to categorise people with mental health problems into more finely graded groups. A medium score on the mental health index (1.51–2.00) can be interpreted as subclinical or subthreshold levels of depression or anxiety disorders, and a high score (2.01–4.00) as a value that may approximate clinical levels. We make this argument because approximately 7% of the participants would be defined as having mental distress if we applied a previously suggested cutoff value (≥2.15) [24], which converges with Norwegian health registry (6.9%) and clinical interview (8.1%) data for depression diagnosis in the general population [33].

## Methodological considerations

A major limitation was the use of a single measurement to assess alcohol intake and mental health problems. Most participants attended the baseline survey in midlife (mean age: 44 years) and a long follow-up period was needed to power the analyses. Changes in alcohol intake or mental health during the follow-up period might have diluted the observed associations. Changes in alcohol intake before a survey, especially 'sick quitters', can give rise to reverse causation through an inflated risk if abstainers are used as the reference category [32]. A predominantly midlife sample, however, might be preferable to an older sample in this regard [34]. We also used low-level drinking as the referent category, and not current abstainers, in order to reduce bias from residual confounding or reverse causation [35]. Previous studies showed more mental distress among abstainers [36], and in a subset we found a higher use of antidepressants and tranquillizers among current abstainers, as well as among people reporting high intake of alcohol and people with higher scores on the mental health index.

Systematic underreporting of alcohol intake was indicated by the higher increase in HDL-C per unit (1 gram/day) increase in self-reported alcohol intake among women (0.013 mmol/l) and men (0.007 mmol/l) in comparison with short-term experimental studies (1 gram/day, approximately 0.0035 mmol/l) [28]. A small difference in slope according to mental health was observed, which seems too small to cause differential bias.

Norway is a country with rather low per capita alcohol consumption [37], a free healthcare system, and a high life expectancy. We argue that the findings in this study can be generalised to a large segment of the general Norwegian population, but there are important limitations. Because of the sampling profile of the health surveys, and the Age 40 Program in particular, the vast majority of the participants in the study population attended a health survey in midlife (73% were between 35 and 50 years of age). A study performed exclusively in younger or older populations, where the nature of non-CVD deaths could differ, may return somewhat different results. A related issue is the possibility that individuals who were the most vulnerable or susceptible to the negative effects of alcohol or mental health problems could have been more likely to have died prior to the health surveys, which could result in underestimation of the risk associated with alcohol intake and mental health problems [38]. The study population also deviated from the general population for 2 reasons. First, the response rate in the health surveys that constituted the source population ranged from 37.5% to 78%. Self-selection into survey attendance likely resulted in a slightly healthier source population than the general population. People with more severe types of mental health problems or more extreme alcohol intake could be underrepresented. Second, the participants excluded for missing values (mostly data on mental health and alcohol intake) were older and more likely to have died during follow-up than the average participant, suggesting that they might have been more frail. The associations could be different, and possibly more pronounced, in more frail or extreme subpopulations, or in countries where average alcohol intake is higher, the drinking pattern different, or mental healthcare less available.

### Implications and future research

This study was performed in a large sample of the general population. The findings suggest that mortality risk is increased in people who report both more mental health problems and a higher alcohol intake, raising the possibility of interaction between risks associated with mental health problems and higher alcohol intake. An implication of this finding might be that the disease burden attributable to mental health problems could be higher in subpopulations that drink more alcohol, and vice versa. Co-addressing mental health and alcohol intake in primary healthcare, perhaps even before problems reach clinical level, may help reduce disease burden attributable to both risk factors by improving quality of life and reducing mortality. The study also provides additional observational data that are in line with the current low-risk drinking guidelines, in which people with mental health problems are advised to consider avoiding or limiting their intake of alcohol. Unfortunately, the study design does not enable us to indicate mechanisms that could be underlying the suggested interaction. A longitudinal study design could provide more answers, especially if it includes repeated measurements over the life course for mental health problems, average alcohol intake (preferably also drinking patterns), and health data related to risk factors, health behaviour, dietary factors, and the use of psychotropic drugs. Such a study could follow the consequences associated with different sequences of events (a high alcohol intake preceding mental health problems or vice versa), both in terms of the risk factors and the mortality risk. Genotyping of large cohorts harbouring relevant information could reveal the role of genetic variants predisposing individuals to both alcohol use and mental health problems over time.

## Conclusions

Our study found that the mortality rates associated with more mental health problems and a high alcohol intake were increased when these risk factors occurred together among people in the general population. Current low-risk drinking guidelines targeted at the general population state that mental health problems are a reason to consider avoiding or limiting alcohol. The study findings are in line with this advice. Clinicians may consider advising patients about the additional harm implicated by this combination even if alcohol intake and mental health problems jointly or alone do not reach clinical levels of severity.

## Supporting information

**S1 Checklist. The Reporting of studies conducted using observational routinely-collected health data (RECORD) guideline, which is an extension of the strengthening the reporting of observational studies in epidemiology (STROBE) guideline.**
(DOCX)

**S1 Fig. Flowchart showing the selection of study participants from the source health surveys and into the study sample.**
(TIF)

**S2 Fig. Crude relationship between alcohol intake and mental health problems as measured by the mean score per question on the mental health index.** The boxplot depicts the mean value of the mental health index with 95% confidence intervals among current abstainers. The curve depicts the smoothed mean of the mental health index according to the average alcohol intake (grams/day) among current drinkers. The grey shading depicts the 95% confidence intervals. The vertical axis has been truncated.
(TIF)

**S1 Protocol. This study is part of a larger research project with a protocol containing information about the planned project, including background, research questions, most of the data sources, main variables, and pre-planned data analysis and study samples.**
(PDF)

**S1 Table. The cardiovascular health surveys constituting the source population (*n* = 307,541 visits).**
(DOCX)

**S2 Table. Descriptive statistics of individuals excluded because of missing values.**
(DOCX)

**S3 Table. Descriptive statistics according to mental health problems and alcohol intake in the study population (Table 1 continued).**
(DOCX)

**S4 Table. All-cause and CVD mortality according to mental health index in the study population overall and stratified by alcohol intake.**
(DOCX)

**S5 Table. All-cause and CVD mortality according to alcohol intake in the study population overall and stratified by mental health index.**
(DOCX)

**S1 Text. Overview of the differences between the study planned per protocol and the study performed.**
(DOCX)

**S2 Text. Programming code for R statistical software: Data cleaning, harmonisation of survey data, and selection of study population.**
(TXT)

**S3 Text. The mental health index.**
(DOCX)

## Acknowledgments

We thank Anneli Pellerud for project coordination and Jon Marius Grasto Wickmann for data management.

## Author Contributions

**Conceptualization:** Eirik Degerud, Gudrun Høiseth, Jørg Mørland, Inger Ariansen, Sidsel Graff-Iversen, Eivind Ystrom, Luisa Zuccolo, Øyvind Næss.

**Data curation:** Eirik Degerud, Inger Ariansen.

**Formal analysis:** Eirik Degerud.

**Funding acquisition:** Øyvind Næss.

**Methodology:** Eirik Degerud, Gudrun Høiseth, Øyvind Næss.

**Project administration:** Eirik Degerud, Øyvind Næss.

**Writing – original draft:** Eirik Degerud.

**Writing – review & editing:** Eirik Degerud, Gudrun Høiseth, Jørg Mørland, Inger Ariansen, Sidsel Graff-Iversen, Eivind Ystrom, Luisa Zuccolo, Øyvind Næss.

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
