## [Decision Letter · Decision Letter 0]

29 Aug 2019

Dear Dr. Degerud,

Thank you very much for submitting your manuscript "Combination of self-reported mental health problems and alcohol intake and the risk of all-cause and cardiovascular disease mortality - A pooled analysis of Norwegian cardiovascular health surveys." (PMEDICINE-D-19-02500) for consideration at PLOS Medicine. 

Your paper was evaluated by a senior editor and discussed among all the editors here. It was also discussed with an academic editor with relevant expertise, and sent to three independent reviewers, including a statistical reviewer. The reviews are appended at the bottom of this email and any accompanying reviewer attachments can be seen via the link below:

[LINK]

In light of these reviews, I am afraid that we will not be able to accept the manuscript for publication in the journal in its current form, but we would like to consider a revised version that addresses the reviewers' and editors' comments. Obviously we cannot make any decision about publication until we have seen the revised manuscript and your response, and we plan to seek re-review by one or more of the reviewers. 

We expect to receive your revised manuscript by Sep 19 2019 11:59PM. Please email us (plosmedicine@plos.org) if you have any questions or concerns.

We look forward to receiving your revised manuscript. 

Sincerely,

Caitlin Moyer, Ph.D.

Associate Editor 

PLOS Medicine

plosmedicine.org

1. Thank you for providing a prospective protocol (S1_protocol). Please note the date the protocol was written, and note in the text any instances where analyses differed from the protocol, and reasons for any such changes.

2. Thank you for the note regarding your data-sharing restrictions. However, PLOS Medicine requires that the de-identified data underlying the specific results in a published article be made available, without restrictions on access, in a public repository or as Supporting Information at the time of article publication, provided it is legal and ethical to do so. Please see the policy at: 

http://journals.plos.org/plosmedicine/s/data-availability

and FAQs at: 

http://journals.plos.org/plosmedicine/s/data-availability#loc-faqs-for-data-policy

In addition, please note that an author cannot serve as primary contact for obtaining the data.

3. Abstract: Please include number of participants included in the study.

4. Abstract and throughout: Please include p-values along with 95% CIs. 

5. Abstract: At Line 51 There is a grammatical issue: “less” should be “fewer”.

6. Abstract: Please move the limitation sentence (last sentence of abstract’s Conclusion section) to be the last sentence of the abstract’s Methods and Findings section.

7. Author Summary: At this stage, we ask that you include a short, non-technical Author Summary of your research to make findings accessible to a wide audience that includes both scientists and non-scientists. The Author Summary should immediately follow the Abstract in your revised manuscript. This text is subject to editorial change and should be distinct from the scientific abstract. Please see our author guidelines for more information: https://journals.plos.org/plosmedicine/s/revising-your-manuscript#loc-author-summary

8. Methods and Results: A table describing cohort demographics (such as table S1) should be found in the main text. 

9. Methods and Results: Please add the following statement, or similar, to the Methods: "This study is reported as per the Strengthening the Reporting of Observational Studies in Epidemiology (STROBE) guideline (S1 Checklist)."

10. Methods and Results: Survey response rates were not included. Please include survey response rates. 

11. Methods and Results: Line 96-98: There appears to be a typo “The Regional Ethics Committee South-East (11/1676) approved the study and gave exemption regarding consent in surveys were this was not obtained.” (were should be where)

12. Methods and Results: Please provide p-values for HRs and CIs reported in Table 1 and Table 2. 

13. Methods and Results: Results from models adjusted for age and sex are presented, as well as multivariable models. Please also present results from unadjusted models.

14. Methods and Results (and supplemental data): Line 332: “Gender” is used, while “sex” is used elsewhere. Both terms are used in the supplemental information. The terms gender and sex are not interchangeable (as discussed in http://www.who.int/gender/whatisgender/en/ ); please use the appropriate term.

15. Discussion: Between the limitations section and conclusions section, please add a brief discussion of implications and next steps for research, clinical practice, and/or public policy.

16. Figures 1, 2, and 3: It is difficult to see the difference between the blue and black dashed lines. Please make this difference more clear (e.g. change the colors, line weights, etc.).

17. Figure S3: For the Kaplan-Meier curve, please provide the number at risk for each time interval.

18. Figure S2, and S3: Please show the axis beginning at zero. If this is not possible, please show a break in the axis.

19. For Figure S3: Please indicate in the figure caption the meaning of the dotted lines vs. solid lines. In the legend of figure S3, p-value = 0 should be p-value < 0.001.

20. For Figure S3: Panel B legend has “Alcohol intake” at the top- remove for consistency, or add a description of different plotted series to graphs in panels A, C, and D.

21. Table S4: The legend has a typo- a period is missing at the end of “HR and 95% CI and p-value derived from Cox models”.

22. References: Please use the "Vancouver" style for reference formatting, and see our website for other reference guidelines: https://journals.plos.org/plosmedicine/s/submission-guidelines#loc-references

23. Thank you for submitting the STROBE checklist. Please confirm that STROBE is the most appropriate reporting guideline, rather than RECORD. Please report your study according to the relevant guideline, which can be found here: http://www.equator-network.org/

Comments from the reviewers:

Reviewer #1: This paper reports findings from an analysis of combined national survey data linked to mortality in which the authors investigated the association between self-reported level of alcohol intake and mortality and the extent to which this varied according to concurrent reported mental health status. The methodological approach is robust, in my opinion, the paper is well-written, and the authors draw appropriate conclusions. My comments are as follows: 

1. Although unlikely in an unselected community sample, I suppose it is possible that the modifying effect is of psychotropic medication rather than mental health status. It would have been helpful to have had some indication of the level of psychotropic medication use in people within the different mental health status groups, even if this couldn't be considered as a covariate. If this is not possible then I feel it needs to be highlighted as a potential limitation. 

2. I found the text a little confusing on what was done about people who had participated in more than one survey. On p10 it is implied that they were included, but it is not clear (beyond data availability) how one survey point was chosen over another. And then on p15 it is implied that participants in multiple surveys were excluded, which seems to contradict the earlier statement. Some clarification of text is needed. 

3. On P16, lines 231-234, it would be helpful if the authors coud clarify for readers what's meant by 'index score' (I assume it is the mental health measure - if so, could it be referred to as this?). 

4. The labelling of different models for rows in Table 1 (p23, lines 315-316) seems to be missing. The same issue applies to Table 2. 

Reviewer #2: This manuscript describes associations of measures of mental health and alcohol consumption with both all-cause mortality and CVD mortality using pooled data from Norwegian health surveys conducted between 1994 and 2002. Linkage to death registration covered mortality up to the end of 2014. The total source population for the study was over 300,000 individuals and over 21,000 had died by 2014 (6,587 from CVD). Cox Proportional Hazards Models were used for the main analyses to describe the independent associations of mental health and alcohol consumption with mortality, and the interaction terms between mental health and alcohol consumption, adjusted first for age and sex only, and then for age, sex, education, marital status, smoking, physical activity, body mass index, resting heart rate, and total cholesterol. The pattern of findings indicated higher mortality at higher bands of mental health index score (all-cause and CVD but less so for the latter); higher mortality for other groups than for the low average alcohol daily intake group (<2 g/day) with increasing mortality at higher levels of current drinking; and an interaction whereby increasing alcohol intake was more strongly associated with increased mortality in groups scoring more highly on the mental health index. This was most evident for the subgroup with high mental health score and high alcohol intake (>=24 g/day) that had very high mortality rates. The patterns were similar when additional covariates were included in the models but the magnitude of differences was attenuated. The pattern of interaction was also shown by expressing the HR per 5g/day increase in consumption within strata of mental health index score, where the HR for the highest stratum was greater than for the two lower mental health strata.

The paper has strengths, especially in terms of the available data and size of population. This facilitates the quantification of the identified interaction. The manuscript also acknowledges limitations, especially the cross-sectional nature of the risk factor data. However, it also presents and discusses the findings in terms of a model of "susceptibility" to the health risks of drinking, where those with more mental health problems have increased susceptibility. This interpretation is the greatest weakness of the manuscript and it evolves from a narrative that is very sparse in terms of providing an account of what mechanisms might underlie the key observations. There are glimpses of what the authors might be thinking, such as the mention in the abstract that mental health problems "might become worse from drinking" but this does not constitute a coherent explanation or hypothesis for future investigation. There is also a statement at lines 172-173 that HDL-C and systolic blood pressure "are likely to play a mediating rather than a confounding role" but it isn't explicit what association/pattern they are mediating. Further, there is a mention of "genetic risk for alcohol addiction" (line 395) in the Discussion. There could well be other plausible accounts of the reported interactions and one possible explanation is that groups with high mental health scores and high current drinking have a longer history of heavy use. Overall, the paper presents an intriguing and potentially useful empirical finding but sheds too little light on what might be going on to bring it about.

In addition to this main point there a number of more specific points for the authors to consider.

1) The mention in the introduction (lines 63-64) of a positive association between alcohol intake and mental health gets into an unnecessary debate. There are a number of studies (not cited) that find poorer mental health in abstainers compared with light drinkers. However, this has very little to do with the ingestion of ethanol* and such J or U shapes (including this study's findings in Table S2) are not a threat to the core analyses of the present research. I note, too, that reference 7 appears to be cited to support both sides of the above debate. (I haven't checked to see whether it does have ambiguous results.)

2) I think more information is needed in the Methods (lines 87-88) about the surveys other than the Age 40 Program rather than readers having to look elsewhere (ref. 16). What age ranges are included? I would be uncomfortable if these other data sets included younger adults where the nature of non-CVD deaths could be very different from those found at older ages. I'm looking for some reassurance here.

3) There were times in the descriptions of the analyses where a particular analysis appeared to be limited to current drinkers but I couldn't always follow why this was so and under what circumstances it was necessary. Current abstainers are included in the tables, and the figures showing the Kaplan-Meier risk curves (Figs. 2 and 3) have lines starting at 0g/day. Can it be made clearer which analyses excluded current abstainers and why?

4) The early parts of the Discussion are quite vague in expressing the quantification of effects. "Increased risk", "more susceptible", "higher" and "considerable" are descriptions which give little idea of whether the differences found are large or small. The Discussion also suffers particularly from the main weakness of the paper as outlined above.

5) The number of participants for the HUBRO Cohort is reported incorrectly in Table S1.

* Lucas, N., Windsor, T. D., Caldwell, T. M. & Rodgers, B. (2010). Psychological distress in non-drinkers: Associations with previous heavy drinking and current social relationships. Alcohol and Alcoholism 45 (1), 95-102.

Bryan Rodgers

Reviewer #3: This paper addresses the important topic of interaction between alcohol use and mental health status on all-cause and CVD-related mortality. However, the paper has several issues that limit the interpretability of the findings:

1. Presentation of methods and results: The statistical methods and results sections were poorly organized and difficult to follow. The authors present stratified smoothed hazard functions (in the figures), hazard ratios for the joint effects of alcohol use and mental health status, Kaplan Meier plots, and log rank tests. It is unclear in the methods and results sections why all of these different types of results are presented (e.g., which hypothesis is addressed by each set of results).

2. Interaction vs effect modification: In the main text, the paper presents joint hazard ratios, which indicate that the authors are interested in interaction between alcohol use and mental health. However, the primary conclusion seems to be that alcohol is worse for people with worse mental health, which seems to indicate that they are interested in describing the different effects of alcohol within strata of mental health status. (in the former case, joint hazard ratios (with a common referent group, are logical to estimate joint effects; in the latter case, one would present hazard ratios for alcohol use within strata of mental health status). The wording of the conclusion should be consistent with the methods and results.

3. Use of nonstandard language: throughout, the papers uses somewhat nonstandard language (e.g., referring to "risks" and "risk differences" to discuss smoothed hazard plots). I recommend staying away from these terms in this context.

Minor comments:

1. Abstract: were expected HRs under interaction based on an assumption of multiplicative or additive interaction

2. Line 82: a figure would be helpful to understand this complex design. As written, it is a bit unclear who the target population is and who the study generalizes to. Are any important groups excluded or overrepresented?

3. Line 113: was the categorization of the index based on the mean score per question?

4. Line 168: Here, or early in the statistical analysis section, I recommend stating clearly which hazard ratios were estimated and why.

5. Line 168: What was the timescale for the hazard ratios?

6. Line 170: Is there a conceptual model or causal diagram that informs which variables are considered mediators vs confounders? I would have expected biomarkers and current health indicators (heart rate, diabetes, serum cholesterol and serum triglycerides) to be on the causal path (ie play a mediating role) as well. 

7. Line 175: The paragraph that starts at this line should clearly state which hazard ratios were estimated to address which scientific hypotheses. As written, it is unclear to me why so many different types of results were presented. For example, why present both the interaction terms with the continuous variables and the joint reference category HRs?

8. Line 181: Please define a, b, and c in this equation. 

9. Line 181: it appears that the authors are evaluating whether the joint HR differs from the expected joint HR under a multiplicative interaction structure. Why was this structure chosen rather than an additive interaction structure? (The authors could have evaluated a departure from additivity using the RERI (see Li R, Chambless L. Test for additive interaction in proportional hazards models. Annals of epidemiology. 2007 Mar 1;17(3):227-36 for details)

10. Line 194: Was any of the missing data described in this paragraph likely to be informative? Seems like those missing alcohol data may have been systematically different from those with alcohol data. 

11. Line 199: what proportion of participants were excluded due to missing data?

12. Line 203: Including a traditional "table 1" with participant characteristics would help readers follow the paper.

13. Line 225: In this section, I recommend interpreting the beta coefficients to guide the reader through the analysis.

14. Line 230: is the mean presented here the average time of follow-up among those who died or overall? This should be better labeled.

15. Line 234: How were expected numbers of deaths computed from Kaplan-Meier curves and log rank tests? This was not described in the methods section and is not a familiar approach.

16. Line 237: the number of CVD deaths was fewer than what? This sentence should be rephrased.

17. Line 241 and throughout: These figures present the smoothed hazard function, which is different from the "risk curve." 

18. Figure 1: Methods for producing these figures should be included in the Methods section. In addition, the rationale for producing these figures is not obvious. What hypothesis is being addressed? What do these figures add?

19. Line 245 and throughout: This line and below lines refer to the "risk difference" but it is unclear where this is coming from. It appears authors are referring to the Figures, but these figures depict a hazard function. Terms like "risk" and "risk difference" have specific definitions in epidemiology and should be avoided in other contexts to avoid confusion.

20. Line 307: what is this measure of interaction and how is it defined? Is this the departure from the expected HR under perfect multiplicativity?

21. Tables: Labels in the first column of tables are unclear

[LINK]

---

## [Decision Letter · Decision Letter 1]

8 Nov 2019

Dear Dr. Degerud,

Thank you very much for submitting your revised manuscript "Combination of self-reported mental health problems and alcohol intake and the risk of all-cause and cardiovascular disease mortality - A pooled analysis of Norwegian cardiovascular health surveys." (PMEDICINE-D-19-02500R1) for consideration at PLOS Medicine. 

Your revised paper was evaluated and discussed among all the editors here, and was sent for re-evaluation by two independent reviewers, including a statistical reviewer (Reviewers 2 and 3). Their reviews are appended at the bottom of this email, and the accompanying reviewer attachment from Reviewer 3 can be seen via the link below:

[LINK]

In light of the review of Reviewer 3, I am afraid that we will not be able to accept the manuscript for publication in the journal in its current form, but we would like to consider a further revised version that addresses the reviewers' and editors' comments. Obviously we cannot make any decision about publication until we have seen the revised manuscript and your response, and we plan to seek re-review by one or more of the reviewers. 

We expect to receive your revised manuscript by Nov 15 2019 11:59PM. Please email us (plosmedicine@plos.org) if you have any questions or concerns.

We look forward to receiving your revised manuscript. 

Sincerely,

Caitlin Moyer, Ph.D.

Associate Editor 

PLOS Medicine

plosmedicine.org

Abstract: Methods and Findings: Lines 44-48: Please revise this sentence to: “For CVD mortality, HRs were 0.93 (0.86, 1.00, p = 0.058), 0.90 (0.76, 1.07, p = 0.225), and 0.95 (0.67, 1.33, p = 0.760) for a low index score combined with light, moderate, and high alcohol intakes, and 1.11 (0.98, 1.25, p = 0.102), 0.97 (0.83, 1.13, p = 0.689), 1.01 (0.71, 1.44, p = 0.956), and 1.78 (1.14, 2.78, p =0.011) for high index score combined with light, moderate, and high alcohol intakes, respectively.” to further clarify which analyses correspond to which values.

Abstract: Background: Lines 27-30: In the final sentence of this section, please add the term “self-reported” (so that it reads “...according to self-reported mental health problems and alcohol intake…”

Abstract: Methods and Findings: Lines 48-51: Please revise this sentence to: “HRs for the combination of a high index score and high intakes (HRs: 2.29 for all-cause and 1.78 for CVD) were 64% (95% CI: 53 – 74%) and 69% (42 – 97%) higher than expected for all-cause mortality and CVD mortality, respectively, under the assumption of a multiplicative interaction structure.” to clarify which analyses correspond to which data values.

Abstract: Methods and Findings: Lines 51-52: Please revise this sentence to: “A limitation of our study is that the findings were based on average reported intakes of alcohol without accounting for the drinking pattern.” or similar to clarify the main limitation of the study for the reader.

Results: Lines 326-328: Please specify which beta-coefficient values correspond to which analyses. Also, please provide a reference to the table where the data are presented showing the relationship between mean mental health index score and alcohol intakes (e.g. Table 1).

Results: Lines 355-357: Please specify which values correspond to which analyses for abstainers, and light, moderate and heavy drinkers.

Discussion: Conclusion: Please expand on your conclusion sentence. (For example, you could also speak to the result of your objective to look at the association between alcohol consumption and CVD/all cause mortality broken down by mental health index, and briefly touch on any important clinical implication.)

S2 Table: Please define the abbreviation “CVD” in the table footnote.

S3 Table: Please define the abbreviations “CVD” and “IHD” in the table footnote.

Figures 1 and 2: Please explain why twice the SD used rather than 95% CIs.

Supporting Information: S1 protocol: Please provide a descriptive title for this item.

Comments from the reviewers:

Reviewer #2: The authors have provided a detailed and clear account of the revisions made to this manuscript in their rebuttal. I have focussed on the general comments I made on the initial submission and the responses to my five specific points. Overall, I will leave the other reviewers to address the revisions made in response to their points but I have checked specific things they raised that were related to my own initial concerns, particularly Reviewers 1's point about psychotropic medication and Reviewer 3's first two points. I am very happy with the responses made by the authors to my original review and find the manuscript to be considerably improved. For the second of my specific points, I agree that little purpose would be served by including the sensitivity analysis with a limited age range in the final manuscript. The proportion of the sample aged under 40 at initial attendance is too small to make a notable difference.

There are some minor issues with the language of the manuscript. Some examples I remember were:

line 283 - "and died more often from all-causes and CVD" would read better as "and more likely to have died from all-causes and CVD";

line 449 - "However, the study design prevent further investigation" should be "However, the study design prevents further investigation";

line 537 - "Such as study" should be "Such a study".

Reviewer #3: Please see attached file.

[LINK]

---

## [Decision Letter · Decision Letter 2]

18 Dec 2019

Dear Dr. Degerud,

Thank you very much for re-submitting your manuscript "Combination of self-reported mental health problems and alcohol intake and the risk of all-cause and cardiovascular disease mortality - A pooled analysis of Norwegian cardiovascular health surveys." (PMEDICINE-D-19-02500R2) for review by PLOS Medicine.

I have discussed the paper with my colleagues and the academic editor and it was also seen again by one reviewer. I am pleased to say that provided the remaining editorial and production issues are dealt with we are planning to accept the paper for publication in the journal.

[LINK]

We look forward to receiving the revised manuscript by Dec 23 2019 11:59PM. 

Sincerely,

Caitlin Moyer, Ph.D.

Associate Editor 

PLOS Medicine

plosmedicine.org

Requests from Editors:

1. Data Availability Statement: Thank you for providing your data availability statement and the contact information for access to your study’s data. Please remove the following text from the statement, as this detail is not needed: “The reason why the data are not made publicly available in a public repository or as a supporting information is because of local legal restrictions as well as ethical restrictions related to privacy. The participants have not consented to their data becoming publicly available and they might lose control of their data, such as their right to have their data deleted.” 

2. Title: Please revise your title according to PLOS Medicine's style. Your title must be nondeclarative and not a question. Please place the study design in the subtitle, after a colon rather than a hyphen. When you revise your title, please update the text of the manuscript as well as the manuscript submission form with your revised title. We suggest, “Association of coincident self-reported mental health problems and alcohol intake with all-cause and cardiovascular disease mortality: A Norwegian pooled population analysis” or similar.

3. Abstract: Methods and Findings: Line 36: Please indicate here that alcohol intake was “self-reported”.

4. Abstract: Methods and Findings: Please provide some summary demographic information for the study participants, e.g. prevalence information, mortality, etc.

5. Abstract: Methods and Findings: Line 42: Please revise to “...light, moderate, and high intakes, respectively.”

6. Abstract: Conclusions: Please revise this sentence to clarify “they” in “...were increased when they occurred together.”

7. Author summary: Please use bullets rather than dashes for separate points.

8. Author Summary: Why was this study done?: Please revise the second point to clarify, we suggest: “Many people both drink alcohol and experience mental health problems, but we do not have much data showing whether the combination of drinking alcohol and mental health problems is associated with additional negative health consequences.”

9. Author Summary: What did the researchers do and find?: Please clarify the second point, we suggest: “The risk of all-cause and cardiovascular disease mortality was higher among people defined with more mental health problems and a high alcohol intake (“>24 g/day”) than would be expected for the linear combination of high alcohol intake and high mental health index.”

10. Author Summary: What do these findings mean?: Please clarify the bullet points and remove causal language, similar to this, according to your intended meaning: 

- The findings suggest that co-occurring alcohol intake and mental health problems are associated with increased negative health effects including all-cause and cardiovascular disease-related mortality.

- Our findings may help to inform clinical recommendations regarding potential risks of alcohol use by individuals with mental health problems.

- The findings warrant future studies with longitudinal data that can shed more light on the mechanisms underlying the interaction between alcohol intake, mental health, and mortality.

11. Introduction: Line 104: Please change “relation” to “relationship”.

12. Introduction: Lines 106-108: The meaning of the word “trait” is not clear, please clarify these sentences.

13. Introduction: Lines 104-111: Please replace “relation” with “relationship”.

14. Introduction: Line 118: Please change “interaction” to “an interaction between mental health problems and alcohol intake.” or similar, according to your meaning.

15. Introduction: Final sentence: Please revise to reduce causal language: “The second objective was to investigate whether risks of all-cause or CVD-related mortality are increased in persons who report mental health problems and alcohol intake in combination, suggesting an interaction between mental health problems and alcohol intake.” or similar

16. Methods: Line 131: Please clarify if by “when attending” you mean “at the time of participation in the survey”

17. Results: Lines 282-289: Please clarify that the 51,754 participants excluded is the total excluded, and you then discuss the breakdown of that number by type of missing data. 

18. Results: Lines 342-343: Please give the minimum follow up time, as well as the maximum.

19. Results: Lines 366-372: Please provide p values for analyses of alcohol intake and the risk of all-cause and CVD mortality.

20. Results: Lines 380-386: Please clarify what is meant when you say “higher” or “higher than expected” by providing the reader with information on what would be “expected” and also describing the HRs you observed, and demonstrating how they differ (with 95% CIs and p values).

21. Results: Lines 406-410: Please provide the 95% CIs and p values for the relationships described here. In the sentence at line 406 please clarify that this describes the relationship for all-cause mortality.

22. Results: Lines 418-420: Please clarify what is meant when you say “test for a difference in slope was positive”.

23. Discussion: Line 540: Please clarify the term “frailer”.

24. Discussion: Line 546-548: Please revise this sentence to avoid causal language, we suggest: “This study suggests that in a large sample of the general population that mortality risk is increased in people who report more mental health problems and a higher alcohol intake, raising the possibility of interaction between risks associated with mental health problems and higher alcohol intake.” or similar.

25. Discussion: Lines 548 and Line 572 (and throughout): Please revise to avoid the use of causal language (“evidence”) as your study is observational and causality cannot be suggested. 

26. Discussion: Conclusions: First sentence: Please add “Our study found…” or similar to the sentence.

27. Discussion: Conclusions: Line 572-574: This sentence is not clear, please revise: “This potentially involves wide population groups extending beyond diagnosed patients with disorders related to alcohol use or mental health.”

28. Figure 2 legend: Please clarify what is meant by “functional forms” in the title.

29. Table 1 Legend: Rather than saying a high score is unfavorable, please clarify specifically what a high score indicates (e.g. indicates greater numbers of self reported mental health issues).

30. Table 2 and Table 3: Please present both the 95% CIs and p values for the interaction effects.

31. S2 Figure: Please clarify in the legend the gray shading represents the 95% CIs.

32. S1 Checklist: Thank you for including the STROBE Checklist. 

33. Please remove the sections (lines 770 and beyond) on Transparency, Competing Interests, Funding, Ethics Approval, and Data Sharing, as this information is automatically pulled together via the manuscript submission system.

Comments from Reviewers:

Reviewer #3: The authors have been very responsive to my previous comments and have now addressed all issues raised. No additional comments.

[LINK]

---

## [Editor Report · Decision Letter 3]

6 Jan 2020

Dear Mr Degerud, 

On behalf of my colleagues and the academic editor, Dr. Charlotte Hanlon, I am delighted to inform you that your manuscript entitled "Association of coincident self-reported mental health problems and alcohol intake with all-cause and cardiovascular disease mortality: A Norwegian pooled population analysis" (PMEDICINE-D-19-02500R3) has been accepted for publication in PLOS Medicine. 

PRODUCTION PROCESS

PRESS

PROFILE INFORMATION

Thank you again for submitting the manuscript to PLOS Medicine. We look forward to publishing it. 

Best wishes, 

Caitlin Moyer, Ph.D.

Associate Editor 

PLOS Medicine

plosmedicine.org